# Adapting Agriculture to Climate Change: A Synopsis of Coordinated National Crop Wild Relative Seed Collecting Programs across Five Continents

**DOI:** 10.3390/plants11141840

**Published:** 2022-07-13

**Authors:** Ruth J. Eastwood, Beri B. Tambam, Lawrence M. Aboagye, Zeynal I. Akparov, Sunday E. Aladele, Richard Allen, Ahmed Amri, Noelle L. Anglin, Rodolfo Araya, Griselda Arrieta-Espinoza, Aydin Asgerov, Khadijah Awang, Tesfaye Awas, Ana Maria Barata, Samuel Kwasi Boateng, Joana Magos Brehm, Joelle Breidy, Elinor Breman, Arturo Brenes Angulo, Marília L. Burle, Nora P. Castañeda-Álvarez, Pedro Casimiro, Néstor F. Chaves, Adelaide S. Clemente, Christopher P. Cockel, Alexandra Davey, Lucía De la Rosa, Daniel G. Debouck, Hannes Dempewolf, Hiba Dokmak, David Ellis, Aisyah Faruk, Cátia Freitas, Sona Galstyan, Rosa M. García, Krishna H. Ghimire, Luigi Guarino, Ruth Harker, Roberta Hope, Alan W. Humphries, Nelissa Jamora, Shakeel Ahmad Jatoi, Manana Khutsishvili, David Kikodze, Angelos C. Kyratzis, Pedro León-Lobos, Udayangani Liu, Ram P. Mainali, Afig T. Mammadov, Norma C. Manrique-Carpintero, Daniele Manzella, Mohd Shukri Mat Ali, Marcelo B. Medeiros, María A. Mérida Guzmán, Tsira Mikatadze-Pantsulaia, El Tahir Ibrahim Mohamed, Álvaro Monteros-Altamirano, Aura Morales, Jonas V. Müller, John W. Mulumba, Anush Nersesyan, Humberto Nóbrega, Desterio O. Nyamongo, Matija Obreza, Anthony U. Okere, Simone Orsenigo, Fernando Ortega-Klose, Astghik Papikyan, Timothy R. Pearce, Miguel A. A. Pinheiro de Carvalho, Jaime Prohens, Graziano Rossi, Alberto Salas, Deepa Singh Shrestha, Sadar Uddin Siddiqui, Paul P. Smith, Diego A. Sotomayor, Marcelo Tacán, César Tapia, Álvaro Toledo, Jane Toll, Dang Toan Vu, Tuong Dang Vu, Michael J. Way, Mariana Yazbek, Cinthya Zorrilla, Benjamin Kilian

**Affiliations:** 1Royal Botanic Gardens, Kew, Wakehurst, Ardingly, Haywards Heath RH17 6TN, UK; cuna44@me.com (R.A.); e.breman@kew.org (E.B.); c.cockel@kew.org (C.P.C.); a.faruk@kew.org (A.F.); u.liu@kew.org (U.L.); j.mueller@kew.org (J.V.M.); t.pearce@kew.org (T.R.P.); m.way@kew.org (M.J.W.); 2Global Crop Diversity Trust, Platz der Vereinten Nationen 7, 53113 Bonn, Germany; beri.bonglim@croptrust.org (B.B.T.); nora.castaneda@croptrust.org (N.P.C.-Á.); hannes.dempewolf@croptrust.org (H.D.); luigi.guarino@croptrust.org (L.G.); nelissa.jamora@croptrust.org (N.J.); matija.obreza@croptrust.org (M.O.); janetoll10@gmail.com (J.T.); benjamin.kilian@croptrust.org (B.K.); 3CSIR—Plant Genetic Resources Research Institute, Bunso P.O. Box 7, Ghana; aboagyelawrencemisa@yahoo.com (L.M.A.); bkwasi16@yahoo.com (S.K.B.); 4Genetic Resources Institute of Azerbaijan NAS, 155 Azadlig Avenue, Baku AZ1106, Azerbaijan; akparov@yahoo.com (Z.I.A.); aydin.asgerov@genres.science.az (A.A.); afig.mammadov@gmail.com (A.T.M.); 5National Centre for Genetic Resources and Biotechnology, Moor Plantation, Ibadan PMB 5382, Nigeria; saladele6083@gmail.com (S.E.A.); okere.anthony@nacgrab.gov.ng (A.U.O.); 6The International Center for Agricultural Research in the Dry Areas, Dalia Bldg, 2nd Floor Bashir El Kassar Street Verdun, Beirut 1108-2010, Lebanon; a.amri@cgiar.org (A.A.); m.yazbek@cgiar.org (M.Y.); 7USDA ARS Small Grains and Potato Germplasm Research, 1691 S 2700 W, Aberdeen, ID 83210, USA; noelle.anglin@usda.gov; 8Estación Experimental Agrícola Fabio Baudrit Moreno, Universidad de Costa Rica, 3 km W of Catholic Church of Barrio San José, La Garita, Alajuela 183-4050, Costa Rica; avillalo2005@hotmail.com (R.A.); nestor.chaves@ucr.ac.cr (N.F.C.); 9Centro de Investigación en Biología Celular y Molecular, Universidad de Costa Rica, Ciudad de la Investigación—C.P., San José 11501-2050, Costa Rica; griselda.arrieta@ucr.ac.cr; 10Malaysian Agricultural Research and Development Institute (MARDI), Persiaran MARDI-UPM, Serdang 43400, Malaysia; akhadijah@mardi.gov.my (K.A.); mshukri@mardi.gov.my (M.S.M.A.); 11Ethiopian Biodiversity Institute, Comoros Street, Yeka Subcity, Addis Ababa P.O. Box 30726, Ethiopia; tesfayeawas@gmail.com; 12Banco Português de Germoplasma Vegetal, INIAV, Quinta de S. José, São Pedro de Merelim, 4700-859 Braga, Portugal; ana.barata@iniav.pt; 13Jardim Botânico, Museu Nacional de Historia Natural e da Ciência, Universidade de Lisboa, R. da Escola Politécnica 56, 1250-102 Lisboa, Portugal; joanabrehm@gmail.com (J.M.B.); maclemente@fc.ul.pt (A.S.C.); 14Lebanese Agricultural Research Institute, Tal Amara, Rayak P.O. Box 287, Lebanon; jbreidy@lari.gov.lb (J.B.); hibadokmak@gmail.com (H.D.); 15Centro de Investigaciones Agronómicas, Universidad de Costa Rica, San José 11501-2060, Costa Rica; arturo.brenes@ucr.ac.cr; 16Embrapa Genetic Resources and Biotechnology, Parque Estação Biológica, Av. W5 Norte (Final), Brasília 70770-917, DF, Brazil; marilia.burle@embrapa.br (M.L.B.); marcelo.brilhante@embrapa.br (M.B.M.); 17Direção Regional do Ambiente e Alterações Climáticas, Rua Cônsul Dabney, Colónia Alemã, Apartado 140, 9900-014 Horta, Portugal; pedro.gp.casimiro@azores.gov.pt; 18Fauna & Flora International, The David Attenborough Building, Pembroke Street, Cambridge CB2 3QZ, UK; alickydavey@gmail.com (A.D.); robertahope@hotmail.co.uk (R.H.); 19Plant Genetic Resources Centre, National Institute for Agricultural and Food Research and Technology (CRF-INIA), CSIC, Finca La Canaleja, A2 km 36, 28800 Alcalá de Henares, Spain; lucia.delarosa@inia.csic.es (L.D.l.R.); rosamaria.garcia@inia.csic.es (R.M.G.); 20Alliance Bioversity International Center of Tropical Agriculture, km 17, Recta Cali-Palmira, Apartado Aéreo 6713, Cali 763537, Colombia; danieldebouck@outlook.com; 21International Potato Center, Avenida La Molina 1895, La Molina, Lima 15023, Peru; davedellis07@gmail.com (D.E.); nmanrique@agrosavia.co (N.C.M.-C.); ricesalas@gmail.com (A.S.); 22Banco de Sementes dos Açores, Rua de São Lourenço, nº 23 Flamengos, 9900-401 Horta, Portugal; catia.f.freitas@azores.gov.pt; 23Institute of Botany after A. Takhtajyan of the National Academy of Sciences of the Republic of Armenia, Acharyan Street 1, Yerevan 0040, Armenia; galstyans@ymail.com (S.G.); annersesyan1@gmail.com (A.N.); papikyanastghik@gmail.com (A.P.); 24National Agriculture Genetic Resources Centre, Nepal Agricultural Research Council (NARC), Khumaltar, Lalitpur P.O. Box. 3605, Nepal; ghimirekh@gmail.com (K.H.G.); mainalism.rp@gmail.com (R.P.M.); deesshrestha@gmail.com (D.S.S.); 25Natural England, Foss House, Kings Pool, 1-2 Peasholme Green, York YO1 7PX, UK; ruthharker@gmail.com; 26South Australian Research and Development Institute, Plant Research Centre, Waite Precinct, Gate 2b Hartley Grove, Urrbrae, SA 5064, Australia; alan.humphries@sa.gov.au; 27Bio-Resources Conservation Institute, National Agricultural Research Centre, Park Road, Islamabad 45500, Pakistan; sajatoi@gmail.com (S.A.J.); ssadar2@gmail.com (S.U.S.); 28Institute of Botany, Ilia State University, 1 Botanikuri str., 0105 Tbilisi, Georgia; mananakhuts@yahoo.com (M.K.); kikodze.david@dzelkva.com (D.K.); 29Agricultural Research Institute, Athalassa, P.O. Box 22016, Nicosia 1516, Cyprus; a.kyratzis@ari.gov.cy; 30Instituto de Investigaciones Agropecuarias, Fidel Oteíza 1956, Pisos 12, Providencia, Santiago 8320000, Chile; pleon@inia.cl (P.L.-L.); fortega@inia.cl (F.O.-K.); 31Independent Consultant, Phoenix, AZ, USA; daniele.manzella@fao.org; 32Institute of Agricultural Science and Technology, km 21.5 Highway to the Pacific, Bárcena, Villa Nueva, Guatemala; mmerida@icta.gob.gt; 33National Botanical Garden of Georgia, 1 Botanicuri Street, 0105 Tbilisi, Georgia; tsirapantsu@yahoo.com; 34Agricultural Plant Genetic Resources Conservation and Research Centre, Agricultural Research Corporation, Wad Medani P.O. Box 126, Sudan; eltahir81@yahoo.com; 35Instituto Nacional de Investigaciones Agropecuarias, Avenida Amazonas y Eloy Alfaro, Edificio MAG, Cuarto Piso, Quito 170518, Ecuador; alvaro.monteros@iniap.gob.ec (Á.M.-A.); marcelo.tacan@iniap.gob.ec (M.T.); cesar.tapia@iniap.gob.ec (C.T.); 36Centro Nacional de Tecnología “Enrique Álvarez Córdova”, km 33.5 Carretera a Santa Ana, San Andrés, Ciudad Arce, La Libertad, El Salvador; aurajdb@yahoo.com; 37Plant Genetic Resources Centre, National Agricultural Research Organization, Plot 2-4 Berkeley Road, Entebbe P.O. Box 40, Uganda; jwmulumba@yahoo.com; 38ISOPlexis—Centro de Agricultura Sustentável e Tecnologia Alimentar, Universidade da Madeira, Campus da Penteada, 9020-105 Funchal, Portugal; miguel.carvalho@staff.uma.pt (M.A.A.P.d.C.); humberto_nobrega@yahoo.com (H.N.); 39Kenya Agricultural and Livestock Research Organisation, Genetic Resources Research Institute, Nairobi P.O. Box 30148-00100, Kenya; desterio.nyamongo@kalro.org; 40Department of Earth and Environmental Sciences, Pavia University, Via Sant’Epifanio 14, 27100 Pavia, Italy; simone.orsenigo@unipv.it (S.O.); graziano.rossi@unipv.it (G.R.); 41CITAB—Centro de Investigação e Tecnologias Agroambientais e Biológicas, 5001-801 Vila Real, Portugal; 42Institute for the Conservation and Improvement of Valencian Agrodiversity (COMAV), Universitat Politècnica de València, Camino de Vera 14, 46022 Valencia, Spain; jprohens@btc.upv.es; 43Botanic Gardens Conservation International, Descanso House, 199 Kew Road, Richmond TW9 3BW, UK; paul.smith@bgci.org; 44Subdirección de Recursos Genéticos, Instituto Nacional de Innovación Agraria, Av. La Molina 1981, La Molina, Lima 15024, Peru; dsotomayor@lamolina.edu.pe; 45Facultad de Ciencias, Universidad Nacional Agraria La Molina, Av. La Molina s/n, La Molina, Lima 15024, Peru; 46Food and Agriculture Organization of the United Nations, Viale delle Terme di Caracalla s/n, 00153 Roma, Italy; alvaro.toledo@fao.org; 47Plant Resources Center, Vietnam Academy of Agricultural Sciences, An Khanh, Hoai Duc, Ha Noi 131000, Vietnam; vdtoannga2003@gmail.com (D.T.V.); tuongvd.prc@gmail.com (T.D.V.); 48Joint FAO/IAEA Centre of Nuclear Techniques in Food and Agriculture, Plant Breeding and Genetics Section, 1400 Vienna, Austria; cinzocis@gmail.com

**Keywords:** plant genetic resources, crop wild relatives, seed collection, ex situ conservation, food security

## Abstract

The Adapting Agriculture to Climate Change Project set out to improve the diversity, quantity, and accessibility of germplasm collections of crop wild relatives (CWR). Between 2013 and 2018, partners in 25 countries, heirs to the globetrotting legacy of Nikolai Vavilov, undertook seed collecting expeditions targeting CWR of 28 crops of global significance for agriculture. Here, we describe the implementation of the 25 national collecting programs and present the key results. A total of 4587 unique seed samples from at least 355 CWR taxa were collected, conserved ex situ, safety duplicated in national and international genebanks, and made available through the Multilateral System (MLS) of the International Treaty on Plant Genetic Resources for Food and Agriculture (Plant Treaty). Collections of CWR were made for all 28 targeted crops. Potato and eggplant were the most collected genepools, although the greatest number of primary genepool collections were made for rice. Overall, alfalfa, Bambara groundnut, grass pea and wheat were the genepools for which targets were best achieved. Several of the newly collected samples have already been used in pre-breeding programs to adapt crops to future challenges.

## 1. Introduction

Ensuring physical and economic access to sufficient, safe, and nutritious food—food security—is a challenge for sustainable agriculture as much today as in the days of Nikolai Vavilov, if not more so. Then as now, numerous factors contribute to the food security challenge, including population growth, income inequality, population distribution, dietary expectations, decreasing land and water for agriculture, and pests and disease threats [1]. In addition, global climate change is predicted to have adverse effects on food production [2,3].

Crop breeding can help to enhance food security, but access to, and sharing of, genetic resources will be required alongside improved strategies and technologies [4]. The International Treaty on Plant Genetic Resources for Food and Agriculture (hereinafter, the Plant Treaty) supports the conservation and sustainable use of plant genetic resources for research and breeding and the fair and equitable sharing of benefits arising from their use [5]. It recognizes the interdependence of countries on crop diversity and has established a Multilateral System (MLS) for Access and Benefit Sharing to ensure that genetic resources of 64 crops are globally accessible in a facilitated manner for food security.

Crop wild relatives (CWR) are genetic resources considered within the Plant Treaty. Although exact definitions vary [6], it is generally agreed that CWR are the closely related species of crops that are found growing in natural habitats and can exchange genetic material with cultivated plants. As wild plants thriving in isolated, extreme, or heterogeneous environments, CWR are expected to have locally adapted alleles that are no longer found in cultivated crops [7]. Therefore, CWR are widely considered to be a valuable source of useful traits and genetic diversity for crop improvement [7,8,9,10,11]. The need to widen the diversity of genepools is particularly evident when large areas planted with uniform, genetically susceptible plants are devastated by outbreaks of pests and diseases [7]; for example, corn leaf blight in the United States in the 1970s [12] and taro leaf blight in the Pacific islands in the 1990s [13]. Tyack and Dempewolf [14] summarized the findings of several studies that estimated the economic value of CWR in crop improvement. The annual benefits attributed to CWR range from USD 8 million to 165 billion, for activities that include providing useful alleles for crop improvement and contributing to the global economy.

Although CWR have been relatively neglected for decades [15], their potential was highlighted by Vavilov, and their use in breeding has been increasing since the 1980s [16]. The use of CWR is widespread and well-established for crops such as barley [17], potato [18], wheat [19,20,21,22] and rice [23,24,25] and developing for others such as alfalfa [26], carrot [27], chickpea [28], eggplant [29,30], finger millet [31], grass pea [32] and sorghum [33,34]. The starting point for use is viable hybridization between the crop and a wild relative. Using traditional breeding methods, it is easiest to make these crosses with species in the so-called primary genepool. It becomes increasingly difficult, often requiring technological intervention, to make crosses with species in the secondary and tertiary genepools. For some crops, even more distant relatives can potentially produce viable crosses [35] and modern methods are opening up access to hitherto unexplored diversity.

Diverse and representative collections of CWR are required to enable research on crop origins and domestication (as pioneered by Vavilov), to search for specific adaptive solutions and to provide sufficient material for base-broadening programs [36,37,38]. In the past, the use of CWR in plant breeding largely focused on resistance to biotic stresses, particularly pest and disease resistance [16]. However, there is now greater recognition of the need to prepare climate-ready varieties that are adapted to local abiotic conditions [10]. “Breeding for climate change” means the improvement of crops through the targeted selection of traits such as tolerance to drought, heat, and salinity, although it may also refer to changes in morphology to improve agronomic processes [11,39,40]. The physiological solutions to adapt to, and mitigate the effects of, climate change may well differ among different crops and regions.

For the potential of CWR to be fully realized in crop breeding, they must be readily available in genebanks and adequately characterized. Responding to this challenge, the global initiative “Adapting Agriculture to Climate Change”, hereinafter the CWR Project [41], was launched with funding from the Norwegian government. The first steps of the CWR Project were the compilation of the Harlan and de Wet CWR Inventory [42] and a global gap analysis [43,44]. These tools allowed the identification and prioritization of the ex situ conservation needs of CWR on a global scale. The same data were also used to investigate in situ conservation priorities for CWR [45]. The project then implemented a coordinated collecting program in 25 countries, sampling the CWR of 28 crops of major global importance; all partner countries agreed to placing the resulting diversity into the MLS of the Plant Treaty.

Collecting CWR for conservation and use in breeding is not a new idea [46] but the large scale of the task is now apparent [43] and the urgency to breed climate-ready crops is unprecedented [47]. In addition, the environments where many CWR grow are vulnerable to changes in climate and land use [48]. Securing CWR ex situ is required to prevent genetic erosion, including local extinction. The CWR Project highlights the importance of international coordination and collaboration in the conservation and use of CWR for research and breeding. This project involves a wide range of partners from all around the world and is motivated by scientifically informed priority setting. The CWR Project has followed in the footsteps of Vavilov, carrying out seed collecting expeditions across five continents [38]. A review of a partial dataset of collections made during this project has been published [49], but here we evaluate the complete collecting effort of the CWR Project.

## 2. Results

### 2.1. Summary of Seed Collection

During 2013–2018, partner organizations collected 4587 unique seed samples from the 25 target genepools across the 25 partner countries (Figure 1, Appendix A). Samples were collected across seven of Vavilov’s originally identified centers of crop origin [46] (Figure 2). The samples were from 27 genera, at least 321 species and at least 355 taxa. Ninety-nine samples are yet to be identified to the species level. At the time of duplication, 85 species were new to the Millennium Seed Bank (MSB). A total of 13 taxa had no previous seed collections in the MLS. Overall, 38 samples were collected for species with the International Union for Conservation of Nature (IUCN) conservation status of Critically Endangered (one species), Endangered (one species), Vulnerable (two species) and Near Threatened (six species; Appendix A).

The number of samples and taxa per genepool are listed in Figure 3. Potato was the most highly collected genepool, with 474 samples, closely followed by eggplant, with 438 samples, both from the genus *Solanum* (Figure 3a). Potato also had the largest number of taxa collected (54 taxa), far more than any other genepool targeted in the project. The least collected genepool was chickpea, with 16 taxa. Rice had the highest number of samples (147) in the primary genepool. All of the Bambara groundnut and carrot samples were in the primary genepool taxa (Figure 3b).

Vetch wild relatives were collected from the largest number of countries (13), while Bambara groundnut wild relatives were only collected in only one country, Nigeria (Appendix A). Partners in Pakistan collected CWR from 17 genepools, while those in El Salvador and Peru concentrated exclusively on bean and potato, respectively.

The number of samples per taxon varied widely. There were fewer than 10 taxon replicates (the same taxa collected in different locations) for 185 taxa. Only 76 taxa were collected more than 20 times. *Sorghum bicolor* subsp. *verticilliflorum* (Steud.) de Wet ex Wiersema & J. Dahlb. was collected 107 times across four countries, with 77 of those samples collected in Sudan.

### 2.2. Collection Metric

We calculated a collection metric for each genepool on the basis of the percentage of samples collected from the initial target list and the percentage of species collected from the initial target lists. The collection metric showed Bambara groundnut to be the genepool for which collection targets (number of collections and taxa) were best achieved, closely followed by alfalfa, grass pea and wheat (Figure 4, Table 1 and Appendix A). Collecting samples in the genepools of bean, chickpea, cowpea, pigeon pea and potato proved most difficult. For example, 60% of the target chickpea species were collected, but only 30% of the planned samples. However, more than 60% of the target species were collected for all the genepools.

### 2.3. Multiplication, Safety Backup and Use

To date, 233 species and 4019 samples of the materials collected by the CWR Project have been further shared from The Royal Botanic Gardens, Kew (hereinafter, Kew) to national and international genebanks for multiplication, safety backup, and use (Table 2).

Through this process, material will in due course be backed up in the Svalbard Global Seed Vault (SGSV). In total, 508 unique samples were deposited by the International Center for Agricultural Research in the Dry Areas (ICARDA) in the SGSV between 2019 and 2021, and 51 potato accessions by the International Potato Center (CIP) in 2021. The University of Costa Rica also deposited 51 rice samples in the SGSV in February 2020.

There have been two very substantial multiplication efforts within the CWR Project. ICARDA initially focused on 748 CWR samples in the genepools of barley, faba bean, grass pea, lentil, wheat, and vetch. A total of 624 samples now have sufficient seeds after three multiplication rounds and are available for distribution. CIP is increasing the availability of 270 wild potato samples by multiplication and characterization. To date, all three goals of having sufficient seeds, being virus free and being taxonomically verified have been achieved for 149 accessions. Additional, smaller-scale multiplications have been undertaken at the Agricultural Research Institute of the Ministry of Agriculture in Cyprus, the Ethiopian Biodiversity Institute, the Instituto Nacional de Investigaciones Agropecuarias in Ecuador, the Lebanese Agricultural Research Institute (in collaboration with ICARDA), the Malaysian Agricultural Research and Development Institute, the Plant Genetic Resources Centre in Uganda and the University of Costa Rica.

While it is too early for all of the collected material to have been evaluated or used in breeding, some newly collected CWR are already being incorporated into pre-breeding programs. For example, all alfalfa CWR samples have been regenerated, characterized, and preliminarily evaluated at the South Australian Research and Development Institute (SARDI) for a range of traits including forage production, habit, flower color, pod coiling and fall dormancy. Some alfalfa materials are now part of a wider evaluation trial currently underway in Australia, Chile, Kazakhstan, and China (Inner Mongolia) [26], and will soon be trialed in Kyrgyzstan, Pakistan, and Tunisia. Three pre-bred lines generated using material collected under the CWR Project were included in these evaluation trials. Two of these lines showed moderate–high cold tolerance.

## 3. Discussion

The CWR Project has demonstrated that it is possible to have a successful international collaboration to collect and conserve germplasm across 25 countries in centers of diversity and other regions on five continents. The cooperation of a wide range of partners has improved the diversity, quantity, and accessibility of collections of CWR of 28 crops of global significance (Figure 5). A total of 4587 samples were collected spanning at least 365 CWR taxa (Figure 3). Though the global results are impressive, the impact at the national scale has arguably been even greater. For example, the CWR holdings in Embrapa, Brazil [50], and the Lebanese Agricultural Research Institute were doubled during the project period. In Nepal, the project expeditions resulted in the first CWR collections to be conserved in that country [51].

Unsurprisingly, some genepools proved easier to collect than others. In most cases, finding the target number of samples was more difficult than meeting the target number of species. When analyzed at the genepool level, three legumes (bean, chickpea, and pigeon pea) had particularly low collection metric scores, yet at least 60% of their species targets were met. This pattern was not reflected in all countries, and the collection metric did not assess additional collections. In many cases, species and samples beyond the target list were collected. For example, the Costa Rican CWR Project partners targeted 49 samples in the bean genepool, but successfully collected 70 samples. Despite having the most collections, potato scored weakly in the collection metric, because there were few replicates of the target species. Despite this, other important collections were made, for example, the collection of *Solanum ayacuchense* Ochoa and *Solanum olmosense* Ochoa from Ecuador and Peru, which were previously entirely missing from ex situ collections. Although wild bananas proved difficult to collect, the project has stimulated the launching of additional collecting programs to improve the conservation and availability of banana CWR [52], so the outlook is hopeful even for this genepool.

Some new records and populations were found by the project partners. The gap analysis did not attempt to predict where new species could be found, and gave only poor indications of where new populations of rare and endemic species might be, due to insufficient data. In Costa Rica, the existence of *Phaseolus microcarpus* Mart. was confirmed and partners went on to identify 13 populations, eight of which have been conserved in ex situ seed collections for the first time [53]. A new species, *Phaseolus angucianae* Debouck & Araya, was also discovered in Costa Rica [54]. In Pakistan, *Cicer macracanthum* Popov was found at an altitude 500 m higher than previously recorded. The first collections of *Vicia pyrenaica* L. were conserved in Spain under the CWR Project. New discoveries included the first records of potato spindle tuber viroid in wild *Solanum* species, adding to knowledge about this disease [55].

The chief strength of the CWR Project has been collaboration at all levels. An inveterate networker, Vavilov would have appreciated this. At least 157 people from different scientific fields, countries, and organizations participated in the collecting effort alone (Appendix A). The expanded network of people interested in CWR means that it is more likely the collected samples will be used in research and in the development of climate-ready crops. Already, additional data, such as germination protocols for new species [56] and characterization data [57], have been generated for some samples. This could accelerate the discovery of new adaptive traits and stimulate research [58,59].

Technical support, training provided to collectors and the distribution of collecting guides prepared by the project have contributed to the success of the collecting effort. Substantial resources were invested in these activities, and the investment has clearly paid off. An independent review of the project commented that “the skills and experience acquired by the national partners has enhanced their technical and project management capacity and positions them for participation in future global initiatives” [60]. For some countries in which existing data were minimal and ecogeographic diversity considerable, the maps of predicted species distributions in the guides were less reliable, but will improve in the future as more data are available and analytical tools evolve. Adequate investment in such supporting elements is strongly recommended for future projects.

Many challenges were encountered from planning to implementation of the project. For example, international access and benefit-sharing laws and phytosanitary regulations, and their national implementation, were often difficult to navigate. Some countries were unable or unwilling to participate due to impediments in implementing the Plant Treaty. Encouragingly, some countries that are not Parties nevertheless agreed to work under the framework of the Plant Treaty for this project. In Georgia, it served as a learning experience prior to the ratification of its membership. Obtaining permits for collection and export took far longer than expected. In some cases, authorizations had to be obtained at a local level and implemented in each planned collecting area. The treatment of CWR varied in different countries. In some cases, they fell under environmental regulations, in other cases the agricultural ministry, and in some cases both.

Alarmingly, collecting reports detailed many instances of habitat destruction and damaging land-use changes. Despite collectors’ best efforts, some under-collected species could still not be found and known populations of species were found to have been destroyed. The last known population of *Aegilops crassa* Boiss in Armenia was not found due to construction in its habitat. In Malaysia, collectors reported that *Solanum cumingii* Dunal had disappeared from its previously known locations, which are now farmland. In 2018, a collector returned to the locality of San Pedro de Pilas, Lima Province, central Peru, where he had collected *Solanum cantense* Ochoa in semi-arid scrubland more than 20 years previously. Lamentably, the scrubland had been replaced by a football field and a water reservoir.

Ironically, collecting CWR, one of the solutions for adapting to, and mitigating the effects of, climate change, is getting harder due to climate change. The Intergovernmental Panel on Climate Change [61] reports that widespread and severe losses and damage to human and natural systems are being driven by human-induced climate change that increase the frequency, intensity, or duration of extreme weather events. Extreme weather conditions indeed complicated collecting efforts, by altering the phenology of flowering and fruiting, limiting seed set and preventing physical access to some populations. In 2014, the Cypriot team started collecting while the country was under one of the worst droughts in its history. In 2016, in West Kordofan state, Sudan, rains were poor, and plants struggled to set seed. Unseasonable weather across Vietnam in 2016 resulted in premature seed maturation. The El Niño rains and flooding in 2017 left roads in Peru impassable and collections had to be postponed until the following year. In Spain, where the spring of 2016 was extremely dry, collections were difficult and sometimes, impossible. Then in 2017, in Andalusia, the vegetative cycle of the plants was altered by unseasonable weather and fruits of *Lathyrus amphicarpos* L. were produced earlier than usual.

A major legacy of the CWR Project will be increased national capacity, knowledge, and passion for CWR collection. The sustainability of the initiative depends on the integration of CWR conservation and use into national priorities, and the availability of national funding for further collecting and maintaining collections. There is evidence of this happening. In Pakistan and El Salvador, CWR collecting work has continued beyond the CWR Project. In Nepal, the collection of CWR has become an additional government-funded activity for the genebank. The University of Costa Rica has funded additional rescue collecting. New initiatives include a collaboration between The Plant Genetic Resources Programme in Pakistan and the National Herbarium to include CWR species, and the development of a banana field genebank by the National Agricultural Research Centre in Nepal. The pre-breeding project [62] funded by the Templeton World Charity Foundation, Inc. and the Biodiversity for Opportunities, Livelihoods and Development (BOLD) Project [63] funded by the Norwegian government build on the CWR Project and plan to extend CWR pre-breeding and evaluation.

Ongoing climate change, the continuing global pandemic, and an uncertain policy environment mean that this may be the last big global germplasm collecting effort. The project has continued the work started in the 1910s by Vavilov and since taken up by many others, but it is not complete. Gaps in collections remain, genetic erosion continues, and the need for diversity is more urgent than ever. The CWR Project has shown that coordinated action can effectively address CWR conservation across genepools. The need for genetic material for crop improvement is only going to increase given all the challenges faced by humankind. Many more crop genepools within and outside Annex 1 need the focus that was provided by this project. Large-scale, coordinated CWR conservation makes an impact, and must continue.

## 4. Materials and Methods

### 4.1. Collecting Approach

The collecting effort encompassed the wild relatives of 28 crops in Annex I of the Plant Treaty, which belong to 25 genepools: alfalfa (lucerne), apple, African rice, Asian rice, Bambara groundnut, banana, barley, bean, bread wheat, carrot, chickpea, cowpea, durum wheat, eggplant (aubergine), faba bean, finger millet, grass pea, lentil, oat, pea, pearl millet, pigeon pea, plantain, potato, rye, sorghum, sweetpotato, and vetch. Three pairs of crops share genepools: African and Asian rice, banana and plantain, and bread and durum wheat, and are referred to in this paper as rice, banana, and wheat. Species nomenclature followed The Harlan and de Wet CWR inventory [42]. With limited resources, the CWR Project had to translate the conservation priorities from the global gap analysis into a global collecting strategy which was as cost-effectively and pragmatically as possible. This meant balancing genepools and geography to capture maximum diversity both within and among taxa. Appendix A details the CWR Project target list.

### 4.2. Partnership Approach

Coordinators from the MSB at Kew and the Crop Trust approached 34 national partners in countries harboring the greatest numbers of CWR of priority genepools [41,42,43].

Initial lists of taxa were provided at the first approach. Where possible, the CWR Project attempted to initiate or strengthen collaborations between agricultural and conservation organizations within countries, as interest in CWR spans both disciplines. The project required compliance with all applicable national and international legislation and policies. Plant Treaty focal points and the Secretariat of the Plant Treaty provided support. Where fruitful, discussions moved to project development, including confirming the taxa list, and finally to implementation of a grant agreement between the Crop Trust and national partner(s). Partners in the following countries finally participated in the project: Armenia, Azerbaijan, Brazil, Chile, Costa Rica, Cyprus, Ecuador, El Salvador, Ethiopia, Georgia, Ghana, Guatemala, Italy, Kenya, Lebanon, Malaysia, Nepal, Nigeria, Pakistan, Peru, Portugal, Spain, Sudan, Uganda, and Vietnam. Thus, CWR Project partners collected materials in at least seven of Vavilov’s eight ‘main centers of origin of cultivated plants’, and at least 14 of the 25 CWR partner countries were visited by Vavilov. Appendix A shows the genepools targeted in each of the 25 partner countries.

### 4.3. Planning

Country-specific CWR collecting guides were compiled to complement the knowledge of local experts. These aided the planning of collecting missions by including locality data from the CWR occurrence database, as well as a map of the known distribution from herbarium data and existing seed collections and another map of the modeled range indicating gaps for collection (Figure 6) [43]. Where possible, an indication of flowering and fruiting time was included. To facilitate identification, descriptions and images were provided. An image of mature seeds and a recommended collection method were provided to maximize the quality of seeds collected. Partners planned their collecting missions using these guides in combination with guidelines from Kew, including technical information sheets [64], and their own expertise and experience. Some partners developed species identification sheets in their own language or carried out additional analyses. In some cases, multiple national teams conducted collecting missions in the field simultaneously at the optimal time for seed collection across wide distances. Collecting teams included taxonomists to support identification in the field. Collecting organizations had to obtain collecting permits and all other documents required by local authorities prior to initiating field work.

Special kits were assembled by Kew for seed collecting, cleaning, and drying, and the kits were then provided to the partners (Appendix A). These were designed to support best practices in seed handling, to maximize the quality and longevity of the collected materials. Alongside equipment, a training program was organized for national partners through regional training courses in Australia, Azerbaijan, Chile, Georgia, Ghana, Malaysia, Spain, Uganda, and Vietnam, as well as training at the MSB and several national facilities. A total of 174 participants, representing all partner countries, received tailored seed collecting and conservation training.

### 4.4. Collecting

National collecting programs were designed to span multiple years due to the geographic coverage required, the need to find new populations and to allow for variation in weather conditions.

At each collection site, partners assessed the size of the CWR population and the level of seed maturity before randomly sampling approximately 20% of mature seeds available. Seeds were collected from at least 50 plants to represent the genetic diversity of the populations adequately [64]. Where this was not feasible, as many plants were sampled as possible.

The aim was that each unique sample (seeds from one taxon at one sample site) should have three elements: (1) The sample should comprise more than 10,000 seeds. For some taxa, it was necessary to sample other plant parts (e.g., tubers); (2) Dried plant specimen(s) to allow for later verification of the identification and for deposit in both a national herbarium and the Kew herbarium; and (3) Passport data following specific CWR Project requirements (Appendix A), which included taxon, locality, and date.

### 4.5. Post-Collection Handling

Post collection, seed samples were cleaned to remove empty or infested seeds and non-seed plant material, and dried. At this stage, partners flagged small or low-quality collections. In addition, samples were accessioned into the collection database and given a unique identifier. A portion of the sample, generally one-third, was placed into long-term conservation in national facilities. Most partners shipped the rest of the sample to Kew under the standard material transfer agreement of the Plant Treaty for duplicate storage and verification of identification. In cases where shipment to Kew was not possible, i.e., due to a very small collection size or legal requirements, other facilities were sought to ensure safety duplication. Samples collected as tubers were duplicated at other facilities and, where possible, grown into plants from which seeds were later collected (in line with genebank workflows, other conservation methods will be applied in the future if seeds cannot be produced). Samples at Kew were further cleaned and dried, if required, and placed in long-term storage at −20 °C. To check viability and facilitate use, germination tests were carried out for each seed sample.

With the agreement of collecting partners, the CWR of each genepool were sent from Kew to national and international genebanks, where the samples can be multiplied to increase seed availability, used for research and pre-breeding, and backed up in the SGSV.

When samples comprised fewer than 10,000 seeds, project partners first tried to re-collect materials from the same population the following year. Where small sample sizes persisted, multiplication was considered to ensure sufficient material for access. To this end, the CWR Project undertook specific multiplication projects with the CIP and ICARDA. Some collecting partners also carried out multiplication, particularly when specific growth conditions were required.

### 4.6. Data

Passport data for collected samples were gathered by the national partners, included in their databases, and shared with Kew for inclusion in the Seed Bank Database as samples were accessioned in the MSB. These data were then shared with Genesys [65]. A CWR Project page in Genesys details all the samples collected under the project that are now conserved ex situ [51].

### 4.7. Collection Metric

The collection metric was calculated to assess whether the samples and taxa within a genepool were collected according to the initial target lists agreed with partners. The metric was the sum of two criteria; the percentage of samples collected from the initial target list, and the percentage of species collected from the initial target lists. Both criteria were scored 0–5 as follows: 1–50% = 0, 51–60% = 1, 61–70% = 2, 71–80% = 3, 81–90% = 4 and 91–100% = 5. The scale of the collection metric ranged from 0 (no samples from the initial list were collected) to 10 (all species and samples from the initial lists were collected). This allowed us to rank the genepools in terms of the success of collection.

## Figures and Tables

**Figure 1 plants-11-01840-f001:**
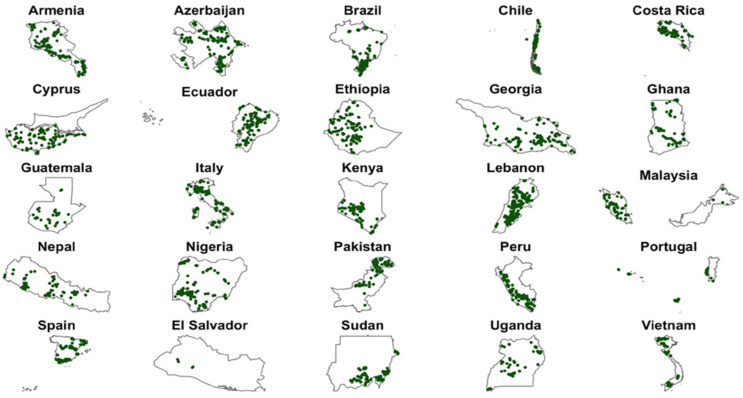
Distribution of unique seed samples collected in the Adapting Agriculture to Climate Change Project (CWR Project). Green dots represent collections made during 2013–2018 in partner countries. The country maps are not to scale.

**Figure 2 plants-11-01840-f002:**
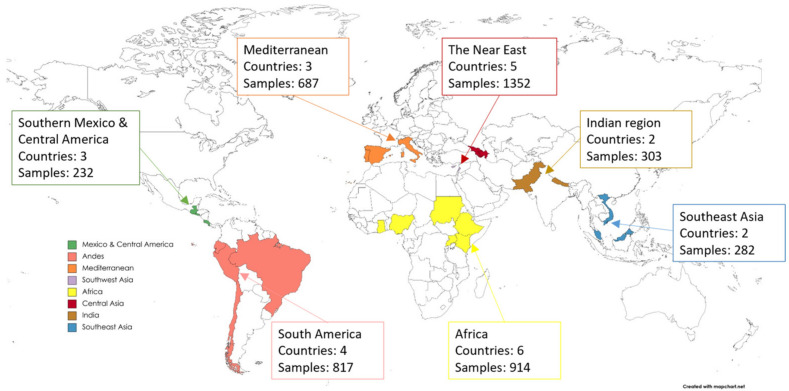
Countries and number of samples collected by the Adapting Agriculture to Climate Change Project (CWR Project) in seven of Vavilov’s centers of origin [46]. Map created using MapChart.net.

**Figure 3 plants-11-01840-f003:**
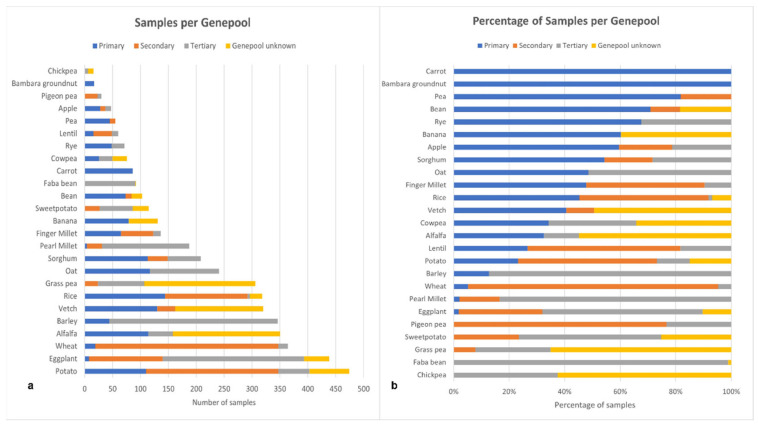
Samples collected per crop genepool categorized by their relationship to the crop. Samples are defined as primary (blue), secondary (dark orange) or tertiary (gray) genepool taxa, or other taxa within the same genus as the crop but whose relationship to the crop is unknown (light orange). (**a**) Number of samples in each category. (**b**) Percentage of samples in each category.

**Figure 4 plants-11-01840-f004:**
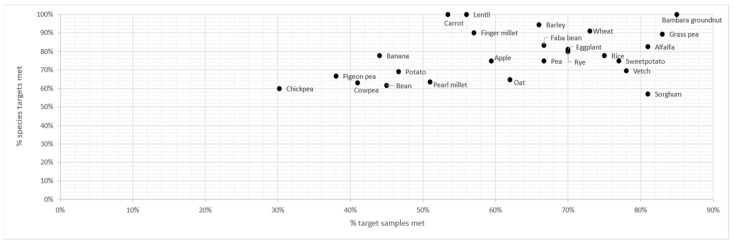
Success of collecting target species and sample numbers per genepool. The collection metric was calculated from these scores.

**Figure 5 plants-11-01840-f005:**
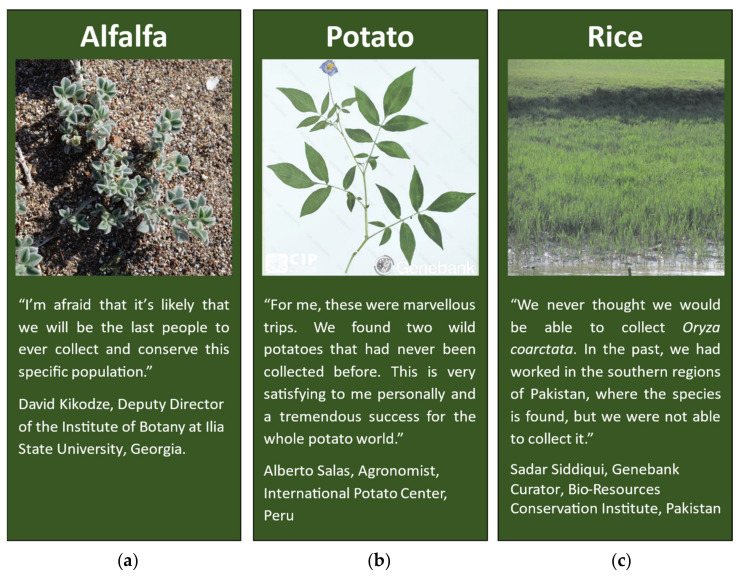
Collectors share their comments on collections of alfalfa, potato and rice. (**a**) Quote refers to *Medicago marina* collected in Georgia. The Georgian Adapting Agriculture to Climate Change Project (CWR Project) partners collected seeds from five populations of *M. marina*. Photo—*Medicago marina*. Ruth Eastwood. (**b**) Quote refers to *Solanum ayacuchaense* and *Solanum ortegae*, which were discovered in the wild for the first time in Peru as part of the CWR Project. Photo—Herbarium specimen of *Solanum ayacuchaense*. International Potato Center (CIP), Peru. (**c**) Quote refers to *Oryza coarctata*, which was collected twice in Pakistan by CWR Project partners. Photo: Wild rice *Oryza coarctata* found in Pakistan. Bio-resources Conservation Institute/Shakeel Jatoi.

**Figure 6 plants-11-01840-f006:**
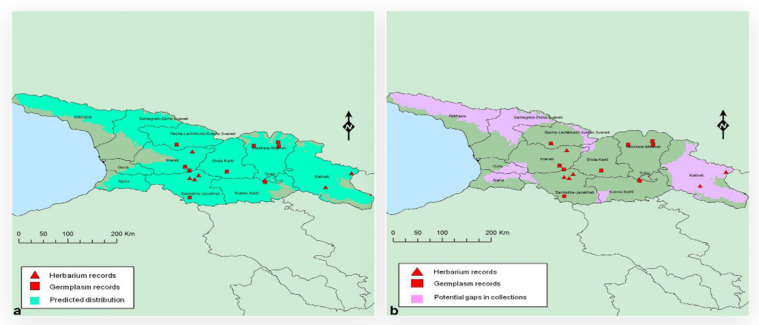
Example of distribution and target maps provided in the Crop Wild Relative (CWR) Collecting Guides. (**a**) Predicted distribution of *Malus orientalis* in Georgia based on herbarium and existing germplasm records. The bright green shaded area, showing the predicted distribution, was modeled using known localities, which are also plotted on the map. (**b**) Target collecting areas (potential gaps) in ex situ collections of *Malus orientalis.* The pink shaded areas highlight areas in the predicted distribution from which no germplasm had yet been collected, indicating where seed collections would be targeted by the CWR Project.

**Table 1 plants-11-01840-t001:** Genepools categorized by collection metric score.

Collection Metric	Genepool
10 All initial species and sample numbers collected	
9	Bambara groundnut
8	alfalfa, grass pea, wheat
7	barley
6	carrot, eggplant, faba bean, lentil, rice rye, sweetpotato
5 Half of initial species and sample numbers collected	finger millet, pea, sorghum, vetch
4	apple, oat
3	banana/plantain, pearl millet
2	bean, cowpea, pigeon pea, potato
1	chickpea
0 No initial species or samples collected	

**Table 2 plants-11-01840-t002:** Shipments of Adapting Agriculture to Climate Change Project (CWR Project) samples from The Royal Botanic Gardens, Kew (Kew) to national and international genebanks for multiplication, use and safety backup. * Material only identified to the genus level has also been shipped.

Crop	Institute	Total Unique Accessions	Total Species
Shipped			
Alfalfa	South Australian Research and Development Institute (SARDI), Australia	348	24
Apple	United States Department of Agriculture (USDA), USA	43	5
Bambara groundnut	International Institute of Tropical Agriculture (IITA), Nigeria	16	1
Banana	International Musa Germplasm Transit Centre (ITC), Belgium	114	7
Barley	International Center for Agricultural Research in the Dry Areas (ICARDA), Lebanon	378	17
Carrot	USDA, USA	83	2
Chickpea	ICARDA, Lebanon	8	3
Cowpea	IITA, Nigeria	60	6
Eggplant	World Vegetable Center (WVC), Taiwan	216	19
Faba bean	ICARDA, Lebanon	75	5
Finger millet	International Crops Research Institute for the Semi-Arid Tropics (ICRISAT), India/Niger	48	9
Grass pea	ICARDA, Lebanon	270	26
Lentil	ICARDA, Lebanon	64	5
Oat	ICARDA, Lebanon	1	1
Oat	Plant Gene Resources of Canada (PGRC), Canada	241	6 *
Pea	USDA, USA	40	2
Pearl millet	ICRISAT, India/Niger	170	22 *
Pigeon pea	ICRISAT, India/Niger	25	3
Rice	International Rice Research Institute (IRRI), Philippines	146	9 *
Rye	Leibniz Institute of Plant Genetics and Crop Plant Research (IPK), Germany	72	6
Sorghum	ICRISAT, India/Niger	195	7 *
Vetch	ICARDA, Lebanon	285	19
Wheat	ICARDA, Lebanon	388	27
**Total**		**3279**	**223**
**Pending**			
Bean	International Center for Tropical Agricultural (CIAT), Colombia	39	9
Eggplant	WVC, Taiwan	126	15
Rice	IRRI, Philippines	96	8
Sweetpotato	International Potato Center (CIP), Peru	21	3
**Total**		**282**	**35**

## Data Availability

Data are available at Genesys (https://www.genesys-pgr.org/subsets/1812d281-0cad-4293-ad71-4d2ef8caabd1 (accessed on 1 April 2021)) for 4264 of the 4587 accessions collected. Of these 4264 accessions, 3544 were accessioned at the MSB, Kew. The remaining 720 accessions could not be transferred to the MSB because of low seed numbers or national restrictions, or because they were sent to CGIAR centers. Data for 323 samples are not available on Genesys because these samples have not yet been accessioned.

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
