# Peer review of "Adapting Agriculture to Climate Change: A Synopsis of Coordinated National Crop Wild Relative Seed Collecting Programs across Five Continents"

_plants, 2022, doi:10.3390/plants11141840_

Round 1

Reviewer 1 Report

This is an excellent article, very timely and an important call to arms for continuation of effort to collect, document and use CWR collections while they can still be found. The article is well written and I commend the lead and other authors for their efforts. I found no obvious mistakes or areas for further improvement.

Author Response

Thank you for your review of our manuscript “Adapting Agriculture to Climate Change: A Synopsis of Coordinated National Crop Wild Relative Seed Collecting Programs Across Five Continents”. We enjoyed reading your comments. Thank you for taking the time to read and think about our paper very carefully.

Best Wishes

Dr. Ruth Eastwood

Reviewer 2 Report

The submited article is not a regular scientific submission, that would be based on experiments and their evaluation. It is a collecting report. However, it is a collecting report summarising several years project devoted to collecting of CWR on all continents, "in footprints of Nikolai Vavilov". The authors declare that the results were partly published and this contribution is an over-all evaluation. It looks like that, because there is much more evaluation and stories than the results. Nevertheless, this type of contribution is good readable, and could atract diversity of readers. It seems very important to have publication that summarises results of extremely wide and large collecting afford, that is important for research and future generations. It does not bring results for basic science, but it extends knowlegge about materials globaly endangered, brings materials that have never been collected to cultivation and genebanks and to potential use by breeders and researchers. The results of the project just come to the time of global changes in environment - the time when threat to rare populations increases. 

I have several comments to the manuscript text and some formal, also marked in manuscript pdf file. 

The title: The article does not deal with adapting of agriculture, it is a collecting report. I recommend shortening of the title to the second part only: A synopsis of coordinated ....

Key words: I recommend to re-order kew words to more logical sequence:

plant genetic resources; crop wild relatives; seed collection; ex situ conservation; food security;  

Intro: no comment, exhaustive

Results: The species collected CWR are nearly all out of Annex I and therefore I doubt they can be within MLS. Please explain. 

Or:  There are no records on their collecting anywhere (in publised sources and PGR databases)?

Graphs: here the genepool concept is mixed with "the same genus" - it should be separated. The sample of the same genus can be GP1 or GP2 or GP3 depending on cross-ability. I would rather see additional another graph with " the same species, the same genus, the same tribe/ subfamily ...

Collection Metric: It would be good to see initial target list (as supplement) otherwise there is no evidence for comparison and support metrics

2.3 Multiplication..: Two of these showed moderate–high cold tolerance.... the info on cold tolerance is only informative if connected to a certain species, otherwise has no sense. Please specify. 

Figure 4.: The photo and text must be in line. Such presentation is for general public and not for researchers. 

left: add picture of M marina or use text for M arborea.

middle: add picture of Solanum ayacuchaense or use text for Solanum commersonii 

4.1 Collecting Strategy: this is not a collecting strategy. It is description of material only,  if you want to keep this chapter/paragraph, include something from methodology of collecting

4.3 Planning: I miss something practical from the planning. It would be good to demonstrate a map as methodical example (citation from your text ....a map of the known distribution from herbarium data and existing seed collections and another map of the modelled range indicating gaps for collection ....) otherwise it is just talking

The sentence is not supported by explanation: Special kits were assembled by Kew for seed collecting, cleaning, and drying, and the kits were then provided to the partners.  what kits? should be explained, shown ,...

4.4 Post-collection handling: ....samples were accessioned into the collection database and given a unique identifier......  are these collecting databases accessibe? where? At least the structure - descriptors should be shown. 

499 Samples collected as tubers were duplicated at other facilities ...... in case of tuber samples the seeds were not available? 

if tubers grown to get seeds - why in vitro was not mentioned or tried?

Conclusion. 

I consider the submission very interesting. The descriptive text should be supported by more results, not only metrics. Maybe description of the most interesting /unique samples with their properties and breeding useful characters. The text can be "a bit more scientific then descriptive. The submission can be published after addressing the commets. 

Author Response

Thank you for your review of our manuscript “Adapting Agriculture to Climate Change: A Synopsis of Coordinated National Crop Wild Relative Seed Collecting Programs Across Five Continents”. Thank you for taking the time to read and think about our paper very carefully. We have used your comments to strengthen our paper.

We have used track changes in MS Word to revise the manuscript.  Below I respond to your useful comments.

  • After careful consideration of all three reviews, we would like to retain the original title. “Adapting Agriculture to Climate Change” was the official CWR Project title and we feel it would be appropriate to include it in this manuscript to highlight the link with the CWR Project.
  • We have followed your advice to re-order the keywords in a more logical sequence.
  • Annex 1: The crops which the CWR Project aimed to support were all included in Annex 1 of the Plant Treaty. The Plant Treaty promotes the exchange of crop wild relatives and other plant genetic resources. For this reason all partner countries agreed to placing resulting diversity into the MLS of the Plant Treaty.
  • Data availability: All of the records for the collections made in the CWR Project can be found in the Genesys PGR database, (https://www.genesys-pgr.org/). As cited in the conclusion Embrapa has published a paper on the CWRP work in Brazil.
  • Figure 3 (previously Figure 2): I have changed the description of the light orange category to “relationship unknown” as suggested by another reviewer.
  • Initial target list: The target list has been added as Supplementary Data S4.
  • Character evaluation/pre-breeding: It is early days in the character evaluation and pre-breeding trials. The alfalfa is the most advanced and the data will be published as soon as possible. By collecting the seeds this project has enabled the possibility for trait assessment, pre-breeding and further breeding in future years. It is the start of the pipeline to improve crops. We are really excited to see what traits are identified and used from the potential we have conserved.
  • Figure 5 (previously Figure 4): Position has been amended and photographs of Medicago marina and Solanum ayacuchaense have been inserted. The caption has been updated to reflect this.
  • Collecting Strategy: We have amended this to “Collecting Approach”.
  • Planning: We have added Figure 6 which shows the maps produced and shared with partners to guide their seed collection work.
  • Collection kits: I have added Supplementary Data S5 which lists the equipment included in the collection kits.
  • Post-collection handling: Each full sample consisted of the seeds, data and a herbarium specimen. Supplementary Method S1 shows the collecting sheet partners followed, this illustrates the data gathered with each sample. Each sample was accessioned into the genebank which collected it and its data placed into that genebank’s database. When sub-samples were shared these data was also shared. Each genebank sent their project data to the PGR database Genesys (https://www.genesys-pgr.org/).  All the data can be viewed in Genesys. Information on how data is handled in Genesys is available online at https://www.genesys-pgr.org/documentation/basics.
  • Tubers samples: We have added more detail to the text to explain that the material will flow into the conservation workflow of the genebank they are in if seed material cannot be obtained. For potatoes, at the International Potato Center (CIP), if growing material for seed collection fails material can be placed in vitro. This can be a lengthy process.

Best Wishes

Dr. Ruth Eastwood

Reviewer 3 Report

Manuscript ID: plants-1788016

Title: Adapting Agriculture to Climate Change: A Synopsis of Coordinated National Crop Wild Relative Seed Collecting Programs Across Five Continents

Authors: Ruth J Eastwood, Beri Bonglim Tambam, Lawrence M Aboagye et al.

REVIEW

This paper presents the results of a large-scale crop wild relative seed collecting missions focused on CWR of 28 crops of global significance for agriculture in 25 countries. It deals with the collected data analysis in a rather straightforward style being sufficiently informative and matching the term synopsis, used in the title. The text is well written and adequately illustrated. I would only suggest considering a few things as detailed below.

As lots of abbreviations (CWR, MLS, MSB, Plant Treaty, etc.) are used throughout the paper, a respective list in the beginning of the paper would be in favor for the readers.

Lines 212–213 (Caption of Figure 1): Adding a note that line maps are presented not within the same scale would be welcome (e.g., the territory of Spain is much larger than that of El Salvador).

Lines 224–225 (Caption of Figure 2): “… or more distantly related taxa but still within the same genus as the crop (light orange)” – what about (simply) unknown GP relationships with the crop, no such taxa?

Lines 233–234: Sorghum bicolor subsp. verticilliflorum (Steud.) de Wet ex Wiersema & J.Dahlb. is a synonym for Sorghum arundinaceum (Desv.) Stapf (see http://www.catalogueoflife.org/annual-checklist/2019/search/all/key/verticilliflorum/fossil/1/match/1 or https://powo.science.kew.org/results?q=Sorghum%20bicolor%20subsp.%20verticilliflorum). It would be good to indicate a reference database used.

With the allusion to the title of the paper, it would be interesting to see a data table or a diagram referring to the collection efforts and achievements by particular continents and/or centers of diversity.

Author Response

Thank you for your review of our manuscript “Adapting Agriculture to Climate Change: A Synopsis of Coordinated National Crop Wild Relative Seed Collecting Programs Across Five Continents”. It was great to read your comments. Thank you for taking the time to read and think about our paper very carefully. We have used your comments to strengthen our paper.

We have used track changes in MS Word to revise the manuscript.  Below I respond to your useful comments.

  • Abbreviations: Taking the editor’s advice these have been added at the end of the manuscript.
  • Figure 1 scale: Note that the scale is not the same for each country has been added to the figure caption.
  • Figure 3 (previously Figure 2): I have changed the description of the light orange category to “relationship unknown”.
  • Reference database: The CWR Project developed and utilized the Harlan and de Wet CWR inventory (https://www.cwrdiversity.org/CWR-Checklist/pages/search/search.php). We have made this clearer in the text.
  • Figure 2 added: This illustrates how the CWR Project worked across the Vavilov centers of origin.

Best Wishes

Dr. Ruth Eastwood